# Values, motivation, and physical activity among Chinese sport sciences students

Yan Liang[1,2☯], Olivier Rascle[1☯], Jian Yang[2☯*], Nicolas Souchon[3☯]

1 Univ Rennes, UR2, VIPS² - UR3646, Rennes, France, 2 Sino-French Joint Research Center of Sport Science, College of Physical Education and Health, East China Normal University, Shanghai, China, 3 Department of STAPS, LICAE, Université de Paris-Nanterre, Nanterre, France

☯ These authors contributed equally to this work.
* yangjianxz@sina.com

## Abstract

Different studies have shown that values and motivation predict physical activity, but no study has tested how values and motivation may interact to predict physical activity. Specifically, the present research aimed to test how values and motivation toward physical activity measured within the SDT could predict global physical activity among Chinese sports science students. The indirect effects of openness to change and self-transcendence values on predicting physical activity through autonomous motivation were significant. These results help us understand how highly abstract psychological constructs such as values may influence physical activity through motivation. Studying values combined with motivation may help better understand the factors that motivate or inhibit physical activity.

## Introduction

Values are highly abstract ideals (e.g., equality, freedom) that influence behaviors across contexts, while motivation is specific to an activity [1]. Specifically, Schwartz's values model (e.g., equality, freedom) is different but related [2,3] to self-determination theory [4], and very few studies have been made on the relation between the two theories [5]. Both autonomous motivation [6,7] and values (e.g., stimulation) tend to predict physical life [8–12]. In the present research, we tested how values and motivation may interact to explain a more active physical life using Chinese sports science students as a sample.

### Self-determined motivation theory (SDT)

SDT attempts to reveal the "reasons" underlying behaviors and suggests that motivation depends on the level of autonomy felt, which is regulated by one's own choices and decisions: i.e., autonomous-intrinsic or controlled-external motivation (see S1 Table). Specifically, the theory focuses on three basic but essential needs in all humans [4].

Satisfying the intrinsic needs for autonomy (e.g., doing an activity with a sense of freedom), affiliation (e.g., doing an activity with friends), and competence (e.g., feeling competent) should enhance autonomous-intrinsic motivation and well-being, while following "extrinsic" aims (e.g., controlling others, wealth, social recognition) or more generally doing an activity

**Data availability statement:** All data files are available from https://doi.org/10.17605/OSF.IO/U5XKW.

**Funding:** This study was supported by the Humanities and Social Science Fund of Ministry of Education of China (Award Number: 22YJA890032), awarded to Jian Yang, Ph.D. The funder, Jian Yang, was responsible for data collection in China.

**Competing interests:** The authors have declared that no competing interests exist.

for external factors such as gaining rewards or avoiding punishment should lead both to a controlled motivation and a lower level of well-being [4].

Autonomous motivation (AM) in physical activity is typically shown through the joy of doing the activity for its own sake and the spontaneous pleasure felt [13]. Meta-analyses indicated that autonomous motivation positively relates to greater physical activity, while controlled motivation has negative or non-significant associations with physical activity levels [6,7].

For instance, Ha and Ng (2015) [14] found that autonomous motivation in Hong Kong middle school students was related to a greater level of objective physical activity, while controlled motivation was not. Similarly, Wang et al. (2016) [15] found that only autonomous motivation positively relates to both self-reported and objective physical activity for Chinese children.

SDT and Schwartz's values are both motivational theories [2,3]. Values are more abstract and general across contexts (e.g., searching for stimulation whatever across context), while autonomous motivation is context-specific: e.g., being intrinsically motivated to play Table Tennis [2].

## Schwartz's model of values

Values are "abstract ideals that act as important guiding principles in one's life" [16]. In the last three decades, Schwartz [16] has provided the most tremendous academic advancement in values research. Shalom Schwartz not only universally identified in its original model 10 categories of values, but also showed how values relate to each other on a circumplex according to their motivational compatibilities and incompatibilities (see Fig 1).

Adjacent values in Schwartz's circumplex are motivationally compatible (e.g., stimulation and hedonism), while opposed values are motivationally incompatible (e.g., security and stimulation). Furthermore, the circumplex is structured around two orthogonal dimensions, each composed of two higher-order values that contrast in their meanings (see Fig 1). Openness to change versus conservation contrasts independence of action, thought, inclination to change against conformity, respect for traditions, and defiance to change. Moreover, self-transcendence versus self-enhancement contrasts an attachment to the welfare of others against an attachment to its own personal interests, dominance, and achievement [3].

This model has been extensively confirmed in more than 100 countries using correlational, experimental, or longitudinal methodologies [17]. Research has shown that values predict attitudes and behaviors across contexts [17], and a few studies have only established a relationship between physical life and values [e.g., 8].

## Values and physical activity

Worsley et al. (2013) [12] measured physical activity through two questions and universalism and conformity values through eight Schwartz items. They found that participants who attached more importance to universalism values tended to be more physically active among the Australian population. In comparison, those who attached more importance to conformity values did not.

Also, Souchon et al. (2015) [11] used the Godin Leisure Time Exercise Questionnaire [18] to quantify physical activity and the Portrait Value Questionnaire [19]. They found that stimulation indirectly predicted physical activity mediated by the attitude toward physical activity among a general French population.

Moreover, with two Polish populations, Skimina et al., in both 2019 and 2021 [9,10], used the Portrait Values Questionnaire (PVQ-RR-57) [3] and the Oregon Avocational Interests Scale (ORAIS) [20] to measure daily behavior, including physical activity. They found in the first study that participants who attached more importance to security-personal and

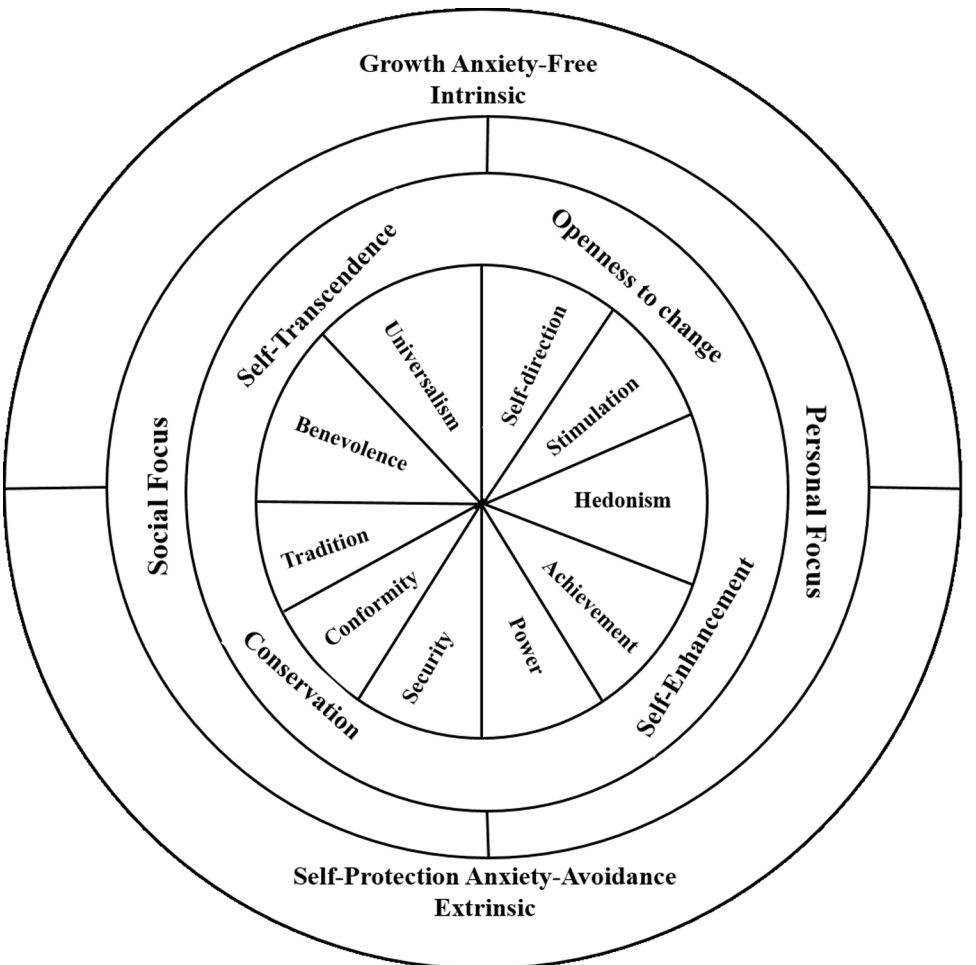

**Fig 1. Schwartz's model of values.**

conformity interpersonal values tended to be less physically active [9]. In the second study, stimulation predicted a higher level of physical activity, while security-personal predicted the opposite [10].

Finally, Liang et al. (2024) [8] used the PVQ-RR-57 to measure values and the IPAQ-long [21] to measure physical activity among French and Chinese sports students. They found that openness to change values predicted more physical activity for French and Chinese sports students, while security and conformity-rule predicted the opposite. Nevertheless, benevolence values only predicted physical activity among Chinese Sports Students.

Consequently, openness to change [8] with stimulation values [10,11]; self-transcendence with both universalism [12] and benevolence values [8], have been shown to predict a larger amount of physical activity. Conversely, conservatism with both security [8], security-personal [10], and conformity interpersonal [9] have been shown to predict the opposite. All of this seems to be very relevant in light of SDT.

## The present study

Schwartz et al. [3] theorize that growth/anxiety-free values: i.e., openness to change and self-transcendence, should be related to autonomous-intrinsic motivation. Specifically,

openness to change encompasses stimulation and self-direction values related to psychological needs for mastery and variety. Self-transcendence encompasses universalism and benevolence values related to needs for affiliation [1–3]. Consequently, individuals prioritizing openness to change and self-transcendence values may focus on satisfying their competence, autonomy, and affiliation needs, while research has shown that satisfying these needs increases autonomous motivation in SDT [4,13,22].

Moreover, self-protection/anxiety-avoidance values: i.e., self-enhancement and conservation, should be related to controlled-extrinsic motivation as self-enhancement values are related to extrinsic needs such as wealth and social recognition (power and part of achievement). Also, conservation values are related to following rules, avoiding social violations (conformity, tradition), or getting protection and attention (security) [1–3]. All these extrinsic needs related to conservation values tend to increase extrinsic motivation in SDT [4,13,23]. Nevertheless, few studies have tested these hypotheses, and little is known about the connection between values, motivation, and behaviors [5,24,25].

Vechionne and Schwartz [5], among 234 Italian high school students, used the PVQ-40 to measure values [19], the Academic Motivation Scale (AMS) [26] to measure autonomous and controlled motivation, and students' grades through teachers' evaluation. They found that self-direction-though values (belonging to openness to change value) predicted school grades mediated through autonomous academic motivation. The more students attached importance to self-direction-though, the more they were intrinsically motivated and the higher they were performant.

Also, Adell et al. [24] tested how human values and autonomous motivation may be related to the intention to make sporting competitions with 115 adolescent basketball players. They used the PVQ-RR-57 [3], the Spanish form of the Behavioral Regulation in Sports Scale [27], and the Future Intention of Practice Scale [28]. They discovered that self-enhancement values predicted a greater intention to practice in the future, mediated through autonomous motivation. The more participants attached importance to self-enhancement values, the less they were intrinsically motivated, and the less they reported an intention to practice sport in the future.

Finally, Balaguer et al. [25] used the Spanish form of the Schwartz Value Survey [29] and the Behavioral Regulation in Exercise Questionnaire [30] with 227 adolescent athletes, but they did not measure physical activity. They found that self-transcendence values were positively and significantly associated with autonomous motivation toward physical exercise participation. Nonetheless, openness to change values unexpectedly was unrelated to motivational constructs in their research.

In the current study, we wished to test the theoretical prediction of Schwartz's value model further, postulating that growth and anxiety-free values (openness and self-transcendence) versus self-protection and anxiety-avoidance values (conservation and self-enhancement) should be more related to autonomous-intrinsic motivation. Specifically, we aimed to test how values and motivation toward physical activity measured within the SDT could predict subjective physical activity among Chinese sports science students.

To our knowledge, numerous studies have been conducted to understand self-determined motivation toward physical activity among general university students in Europe [31,32] or in China [33,34]. Still, very few studies have been undertaken to understand physical activity among sports sciences students [5,35]. Research has shown sports science students doing higher physical activity in Western Europe compared to the general exerciser [35], suggesting sports sciences students pursuing a deep autonomous motivation to do physical activity in this individualistic culture [36]. In contrast, in the Eastern collectivist cultural context, particularly in China, motivation to exercise among general students is heavily influenced by external

pressures like norms, institutional policies, and parental and coach expectations [37–39]. All in all, this makes the choice to study the relation between values, autonomous motivation, and physical activity among sports sciences students in China interesting.

Importantly, one recent study compared the relationship between values and physical activity with Chinese and French sports science students [8]. Results revealed both similarities and differences across countries on values-behaviors relationships. Specifically, openness to change was the only higher-order dimension that predicted a larger amount of physical activity across the two samples. This is interesting as openness to change values should predict autonomous motivation, and research has shown that intrinsic-autonomous motivation positively predicted self-reported and objective physical activity in China [15,40,41]. Also, benevolence values only predicted physical activity in China and not in France [8], while theoretically, self-transcendence values should be related to autonomous motivation [1,2].

Our general hypothesis was that autonomous and controlled motivation should mediate the relations between higher-order values and global physical activity. Our specific hypotheses were:

H1: Openness to change (Independent Variable, IV) and self-transcendence values (IV) would positively predict physical activity (Dependent Variable, DV), and autonomous motivation (Mediator, M) would mediate this relationship. Precisely, a greater attachment to openness to change (IV) and self-transcendence values (IV) would predict a greater autonomous motivation (M), which would predict a greater level of physical activity (DV).

H2: Conservation (IV) and self-enhancement values (IV) would negatively predict physical activity (DV), and autonomous motivation (M) would mediate this relationship. Precisely, a greater attachment to conservation (IV) and self-enhancement values (IV) would predict a lower level of autonomous motivation (M), which would predict a lower level of physical activity (DV).

H3: Openness to change (IV) and self-transcendence values (IV) would positively predict physical activity (DV), and controlled motivation (M) would mediate this relationship. Precisely, a greater attachment to openness to change (IV) and self-transcendence values (IV) would predict a lower level of controlled motivation (M), which would predict a higher level of physical activity (DV).

H4: Conservation (IV) and self-enhancement values (IV) would negatively predict physical activity (DV), and controlled motivation (M) would mediate this relationship. Precisely, a greater attachment to conservation (IV) and self-enhancement values (IV) would predict a higher level of controlled motivation (M), which would predict a lower level of physical activity (DV).

## Materials and methods

### Participants and procedure

A power analysis via G*Power 3.1.9.7 [42] indicated that a whole sample size of 132 is requested to reveal a small effect size of f = .15 with a power of .95 and α = .05 in a liner multiple regression fixed model with five predictors in total. Though earlier investigations [24,25] discovered small and medium effect sizes, we anticipated a small effect size to be cautious.

The study was conducted from April 1 to May 1, 2023, with participants recruited from the Physical Education College of a university in Shanghai, China. Recruitment was conducted after a course session, during which the experimenters explained the study's purpose, content, and relevant procedures. Participation was entirely voluntary. They can choose whether or not to take part and were informed that they can quit at any time.

For those who consented to participate, appointments were scheduled to complete the questionnaire. Surveys were administered in a quiet classroom setting with groups of 15 participants per session. Before the survey, each participant signed an informed consent form outlining the study's purpose, authorized use of subject information, subject confidentiality, and privacy regulations. Upon providing consent, participants proceeded with the formal questionnaires.

To ensure confidentiality, each participant was assigned a unique code containing no identifying information. The informed consent and the questionnaires were securely stored in a locked cabinet within the laboratory, accessible only to the research team.

All actions completed in this research agreed with the ethical standards of the 1964 Helsinki Declaration and its subsequent revisions and with the American Psychological Association (2017) and the British Psychological Society (2009). Furthermore, this research got ethical consent from the University Committee on Human Research Protection of East China Normal University (HR 725-2021).

Overall, 391 participants were recruited, but six were excluded for missing data in physical activity (they did not report any response), and six were excluded for problems in value measurement (they responded to the PVQ-21 with more than five missing responses or provided identical answers to over sixteen value items) [43].

Finally, our sample included 379 sports science college students ($M_{age}$ = 20.560, $SD$ = 1.244; $M_{BMI}$ = 22.214, $SD$ = 6.255) with 280 male participants ($M_{age}$ = 20.730, $SD$ = 1.272; $M_{BMI}$ = 22.790, $SD$ = 7.088) and 89 female participants ($M_{age}$ = 20.100, $SD$ = 1.035; $M_{BMI}$ = 20.586, $SD$ = 2.064).

## Measures

**Physical activity behavior.** We needed to shorten our time measure to increase the number of participants agreeing to do our research. To this end, we used the International Physical Activity Questionnaire short form [21]. The IPAQ-SF is widely used to assess global physical activity and has been extensively validated across cultural contexts [44,45]. Notably, the seven-item short version of the IPAQ has been typically recommended as a cost-effective method to access global PA [44]. Its brevity minimizes the burden on participants, which is particularly advantageous in studies involving multiple measurements. Above all, we employed the well-validated Chinese version of the IPAQ-SF [46].

The IPAQ short assesses physical activity undertaken across a comprehensive set of domains in daily life through 7 items and three different intensities in global physical activity. The first question for measuring vigorous physical activity is: "During the last 7 days, how many days did you do vigorous physical activities like heavy lifting, digging, aerobics, or fast bicycling?". The second question for measuring moderate physical activity is: "During the last 7 days, how many days did you do moderate physical activity like carrying light loads, bicycling at a regular pace, or doubles tennis? Do not include walking". The third question for measuring "walking" is: "Think about the time you spent walking in the last 7 days. This includes at work and home, walking to travel from place to place, and any other walking you have done solely for recreation, sport, exercise, or leisure". For each of these three questions, participants answered the frequency in the last 7 days of vigorous and moderate physical or walking activities and further answered the time they spent on one of those days.

The outcome is a continuous score reported as the median Metabolic Equivalent of Task (MET)-minutes, which express the energy cost of an activity. Median MET values can be calculated for walking (W), moderate-intensity activities (M), and vigorous-intensity activities

(V) using the subsequent methods: walking MET-minutes/week = 3.3 * walking minutes * walking days; moderate MET-minutes/week = 4.0 * moderate-intensity activity minutes * days; vigorous MET-minutes/week = 8.0 * vigorous-intensity activity minutes * days. A combined "total" physical activity MET-min/week can be calculated as the aggregate of walking + Moderate + Vigorous MET-min/week scores.

**Personal value.** Participants completed the Chinese version [47] of the Portrait Value Questionnaire. The PVQ-21 measures the ten original categories of value types with two or three items by category (see more information in Supporting information). PVQ-21 tends to be more valid for the four higher-order values [48].

**Motivation to physical activity.** Participants completed the Chinese form [49] of the Behavioral Regulations in Exercise Questionnaire 2 (see S1 File). Specifically, we used in the present research autonomous (the mean of the identified regulation and intrinsic regulation, $r = .614$, $p < .01$) and controlled motivation (the mean of external regulation and introjected regulation, $r = .099$, $p = .054$).

**Control variables.** Participants indicated their age, gender, Body Mass Index (BMI = kg/m²) through their weight (in kilograms) and height (in meters), and competition level (no competition, local, regional, national, or international) as control variables. Specifically, research has shown that students with higher BMIs report lower levels of autonomous motivation for engaging in physical activities [50], body image concerns and discomfort during exercise [51], or a lower level of endurance during physical activity tasks [52]. Moreover, research has shown that sporting competition levels impact physical activity, as students who do a sport in competition tend to be more physically active than students doing physical activity without sporting competition [8]. Different external pressures (e.g., coaches, head of the sporting club) may be directed toward students doing sports in competition, raising their level of controlled motivation [25].

## Data cleaning and analysis

Our data were analyzed using multiple regression analysis and structural equation modeling (SEM). Following Hayes's Macro Process via bootstrapping method [53], we considered a mediator has a mediational effect when we get simultaneously (a) the indirect effect (IE) of the higher-order value on physical activity via autonomous or controlled motivation (i.e., IE = path a x path b; a = the effect of the higher-order value on the mediator of autonomous or controlled motivation, b = the effect of autonomous or controlled motivation on physical activity) and (b) the bias-corrected 95% CI around the IE from 5000 bootstrap re-samples. We admitted the IE as statistically significant only if its bias-corrected 95% CI excluded zero (see supplementary file for more information).

## Results

### Regression analysis

Table 1 describes the effects of higher-order values on global physical activity and autonomous and controlled motivation. A greater attachment to openness to change predicted a higher global physical activity. Also, a greater attachment to openness to change and self-transcendence values indicated a more important autonomous motivation. Additionally, a higher importance given to self-enhancement values predicted a higher controlled motivation.

Table 2 tests whether autonomous and control motivation predicted global physical activity. Results indicated that greater autonomous motivation predicted greater physical activity, but controlled motivation did not.

**Table 1. Regressions testing the impact of higher-order values on physical activity, autonomous and controlled motivation.**

| | Global physical activity | | | | Autonomous motivation | | | | Controlled motivation | | | |
|---|---|---|---|---|---|---|---|---|---|---|---|---|
| | T | p | B | 95% CI | T | p | B | 95% CI | T | P | B | 95% CI |
| | $R^2 = .034$, $F(8,370) = 2.66$, $p = .008$ | | | | $R^2 = .194$, $F(8,370) = 12.41$, $p < .001$ | | | | $R^2 = .044$, $F(8,370) = 3.16$, $p = .002$ | | | |
| Self-transcendence (raw) | −.21 | .833 | −.02 | [−.15, .12] | 2.92 | .004 | .18 | [.06, .31] | .31 | .759 | .02 | [−.11, .16] |
| Self-enhancement (raw) | .28 | .779 | .02 | [−.10, .13] | 1.14 | .257 | .06 | [−.04, .16] | 4.23 | <.001 | .24 | [.13, .36] |
| Openness to change (raw) | 2.63 | .009 | .16 | [.04, .28] | 4.31 | <.001 | .24 | [.13, .35] | −1.43 | .153 | −.09 | [−.20, .03] |
| Conservation (raw) | −.45 | .653 | −.03 | [−.16, .10] | −.99 | .323 | −.06 | [−.18, .06] | −1.62 | .107 | −.11 | [−.24, .02] |
| Gender | 1.11 | .267 | .06 | [−.05, .28] | 5.07 | <.001 | .26 | [.16, .35] | 1.55 | .122 | .09 | [−.02, .19] |
| Age | −2.05 | .041 | −.11 | [−.21, −.01] | −.16 | .872 | −.01 | [−.10, .09] | 1.59 | .112 | .08 | [−.02, .19] |
| BMI | 1.88 | .061 | .10 | [−.01, .20] | −1.40 | .163 | −.07 | [−.16, .03] | −.84 | .400 | −.04 | [−.14, .06] |
| Competition level | 1.90 | .059 | .10 | [−.01, .20] | −2.34 | .020 | −.11 | [−.21, −.02] | 1.18 | .241 | .06 | [−.04, .16] |
| | $R^2 = .022$, $F(5, 373) = 2.68$, $p = .021$ | | | | $R^2 = .071$, $F(5, 373) = 6.81$, $p < .001$ | | | | $R^2 = .005$, $F(5,373) = 1.35$, $p = .241$ | | | |
| Self-transcendence (center) | −1.13 | .258 | −.06 | [−.16, .04] | .87 | .384 | .04 | [−.06, .14] | −.54 | .593 | −.03 | [−.13, .07] |
| Gender | .76 | .450 | .04 | [−.07, .15] | 4.33 | <.001 | .23 | [.13, .33] | 1.67 | .095 | .09 | [−.02, .20] |
| Age | −1.97 | .050 | −.10 | [−.21, .00] | −.02 | .987 | −.001 | [−.10, .10] | 1.37 | .170 | .07 | [−.03, .18] |
| BMI | 1.95 | .051 | .10 | [−.00, .20] | −.93 | .354 | −.05 | [−.15, .05] | −.59 | .556 | −.03 | [−.13, .07] |
| Competition level | 1.80 | .073 | .10 | [−.01, .20] | −2.51 | .013 | −.13 | [−.23, −.03] | 1.20 | .230 | .06 | [−.04, .17] |
| | $R^2 = .019$, $F(5, 373) = 2.44$, $p = .034$ | | | | $R^2 = .072$, $F(5,374) = 6.85$, $p < .001$ | | | | $R^2 = .051$, $F(5,373) = 5.09$, $p < .001$ | | | |
| Self-enhancement (center) | −.34 | .733 | −.02 | [−.12, .08] | −.97 | .334 | −.05 | [−.15, .05] | 4.32 | <.001 | .22 | [.12, .32] |
| Gender | .85 | .396 | .05 | [−.06, .15] | 4.34 | <.001 | .23 | [.13, .33] | 1.45 | .149 | .08 | [−.03, .18] |
| Age | −2.11 | .035 | −.11 | [−.21, −.01] | .01 | .990 | .001 | [−.10, .10] | 1.64 | .103 | .08 | [−.02, .19] |
| BMI | 2.04 | .042 | .11 | [.00, .21] | −.93 | .351 | −.05 | [−.15, .05] | −.80 | .422 | −.04 | [−.14, .06] |
| Competition level | 1.92 | .055 | .10 | [−.00, .21] | −2.53 | .012 | −.13 | [−.23, −.03] | .99 | .321 | .05 | [−.05, .15] |
| | $R^2 = .030$, $F(5,373) = 3.31$, $p = .006$ | | | | $R^2 = .087$, $F(5,373) = 8.24$, $p < .001$ | | | | $R^2 = .006$, $F(5,373) = 1.460$, $p = .202$ | | | |
| Openness to change (center) | 2.09 | .038 | .11 | [.01, .21] | 2.70 | .007 | .14 | [.04, .23] | −.90 | .368 | −.05 | [−.15, .06] |
| Gender | 1.14 | .257 | .06 | [−.05, .17] | 4.68 | <.001 | .25 | [.14, .35] | 1.56 | .120 | .09 | [−.02, .20] |
| Age | −2.07 | .039 | −.11 | [−.21, −.01] | .12 | .902 | .01 | [−.09, .11] | 1.31 | .191 | .07 | [−.04, .17] |
| BMI | 2.04 | .043 | .10 | [.00, .21] | −1.00 | .318 | −.05 | [−.15, .05] | −.56 | .579 | −.03 | [−.13, .07] |
| Competition level | 1.82 | .070 | .10 | [−.01, .20] | −2.75 | .006 | −.14 | [−.24, −.04] | 1.30 | .195 | .07 | [−.04, .17] |
| | $R^2 = .029$, $F(5,373) = 3.29$, $p = .006$ | | | | $R^2 = .100$, $F(5,373) = 9.45$, $p < .001$ | | | | $R^2 = .010$, $F(5, 373) = 1.73$, $p = .127$ | | | |
| Conservation (center) | −2.05 | .041 | −.11 | [−.21, −.00] | −3.58 | <.001 | −.18 | [−.28, −.08] | −1.46 | .146 | −.08 | [−.18, .03] |
| Gender | 1.20 | .231 | .07 | [−.04, .17] | 4.95 | <.001 | .26 | [.16, .37] | 1.96 | .051 | .11 | [−.00, .22] |
| Age | −1.98 | .048 | −.10 | [−.21, −.00] | .29 | .775 | .01 | [−.08, .11] | 1.41 | .159 | .07 | [−.03, .18] |
| BMI | 1.95 | .052 | .10 | [−.00, .20] | −1.15 | .251 | −.06 | [−.15, .04] | −.67 | .538 | −.03 | [−.13, .07] |
| Competition level | 1.95 | .052 | .10 | [−.00, .21] | −2.59 | .010 | −.13 | [−.23, −.03] | 1.28 | .200 | .07 | [−.04, .17] |

## Mediation analysis

The suggested model was tested via Structural Equation Modeling. After controlling for gender, age, competition level, and BMI, the model was close to a good fit to the data, χ2/df = 2.626, NFI = .877, IFI = .920, CFI = .916, and RMSEA = .066. The factor loadings and path coefficients are presented in Fig 2.

Results indicated that two indirect effects were significant (see Table 3). The indirect effect of openness to change values in predicting physical activity through autonomous motivation is significant ($B = .03$, $SE = .01$, 95% CI [.01, .06], $p = .016$), with the bias-corrected 95% CI excluding zero. Also, the indirect effect of self-transcendence values in predicting physical activity through autonomous motivation is significant ($B = .02$, $SE = .02$, 95% CI [.01, .06],

**Table 2. Regressions testing the impact of motivation on global physical activity.**

| | Global physical activity | | | |
|---|---|---|---|---|
| | T | *p* | *B* | 95%CI |
| *R²* = .050, *F*(9,369) = 3.23, *p* = .001 | | | | |
| Intrinsic regulation | .24 | .808 | .02 | [−.12, .16] |
| Identified regulation | 1.79 | .074 | .14 | [−.01, .29] |
| Introjected regulation | −1.48 | .140 | −.09 | [−.21, .03] |
| External regulation | 1.20 | .233 | .08 | [−.05, .22] |
| Amotivation | −2.22 | .027 | −.16 | [−.30, −.02] |
| Gender | .53 | .597 | .03 | [−.08, .14] |
| Age | −2.03 | .043 | −.11 | [−.21, −.00] |
| BMI | 2.14 | .033 | .11 | [.01, .21] |
| Competition level | 2.22 | .027 | .12 | [.01, .22] |
| *R²* = .041, *F*(6,372) = 3.68, *p* = .001 | | | | |
| Autonomous motivation | 2.97 | .003 | .16 | [.05, .26] |
| Controlled motivation | −1.32 | .189 | −.07 | [−.17, .03] |
| Gender | .29 | .776 | .02 | [−.09, .12] |
| Age | −2.04 | .043 | −.11 | [−.21, −.00] |
| BMI | 2.16 | .031 | .11 | [.01, .21] |
| Competition level | 2.39 | .018 | .13 | [.02, .23] |

*p* = .022), with the bias-corrected 95% CI excluding zero. Moreover, we further tested the separate mediation effects on autonomous, controlled, and a-motivation in different single models (See S1 File).

## Discussion

We aimed to study how values and motivation may be interrelated in predicting physical activity among Chinese sports science students. Our first hypothesis was that a greater attachment to openness to change (IV) and self-transcendence values (IV) would predict a greater autonomous motivation (M), which should predict a greater level of physical activity (DV). The results were in agreement with this hypothesis. Results indicated that openness to change and self-transcendence values predicted autonomous motivation, which in turn predicted a greater amount of physical activity. These results obtained in China are in agreement with a recent finding showing that both openness to change and benevolence values predicted physical activity in China [8], and with studies showing that autonomous motivation predicted physical activity in China [15,40,41] or in different countries [6,7].

These results are interesting in understanding the connections between values and SDT. They confirm Schwartz's [1] and Maio's [2] expectations and match with earlier research analyzing the relations between Schwartz's values and SDT [5,22,23]. Specifically, Vechionne and Schwartz [5] found that the more students in high school attached importance to self-direction-though, the more they were intrinsically motivated and the higher they were per-formant. Also, Balaguer et al. [25] found that self-transcendence values were related to more autonomous motivation toward physical activity.

These results are also interesting to better understand the values and physical activity relationships. We included autonomous and controlled motivation as potential media-tors to strengthen our understanding of processes explaining connections between values

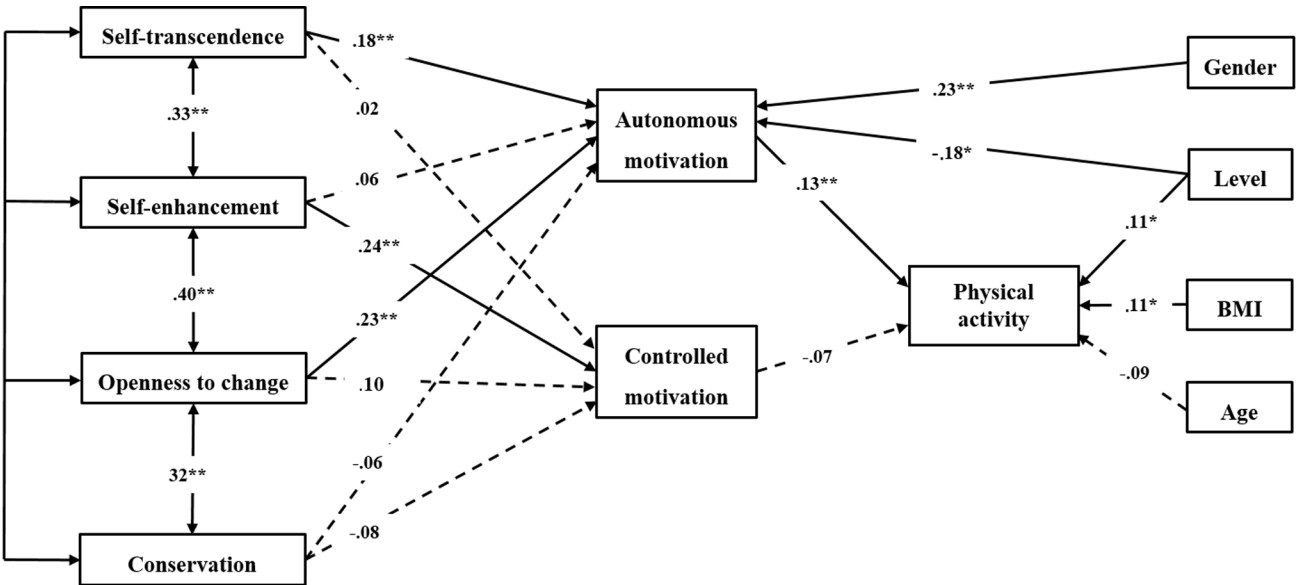

*Note.* Solid lines represent the significant correlations and the dashed lines represent the non-significant correlation, * means *p*<.05, ** means *p*<.01

**Fig 2. Tested model in the present study.**

**Table 3. Mediation analysis of autonomous/controlled motivation between four higher values and physical activity.**

| Indirect effect | B | p | SE | 95%CI |
|---|---|---|---|---|
| Self-transcendence → AM → PA | .03 | .006 | .01 | [.01, .06] |
| Openness to change → AM → PA | .04 | .001 | .02 | [.02, .07] |
| Self-enhancement → AM → PA | .01 | .243 | .01 | [.00, .03] |
| Conservation → AM → PA | −.01 | .263 | .01 | [−.03, .01] |
| Self-transcendence → CM → PA | .00 | .589 | .01 | [−.02, .01] |
| Openness to change → CM → PA | .01 | .173 | .01 | [.00, .03] |
| Self-enhancement → CM → PA | .02 | .222 | .02 | [−.04, .01] |
| Conservation → CM → PA | .01 | .195 | .01 | [.00, .03 |

and physical activity. Important values influence how individuals perceive, interpret, and understand situations [17]. Attaching importance in China to openness to change and self-transcendence values may motivate individuals in situations in which they do physical activity (e.g., sporting clubs or simply in leisure time), to generate opportunities to learn or develop connections with others. The central motivations in China leading to autonomous motivation could be making friends, being in harmony with others [8], or developing skills rather than being dominant. Chinese individuals attaching importance to these higher-order values may also interpret physical activity situations as allowing them to experiment with freedom, for example, through running or cycling (e.g., mountain bike) in nature.

Moreover, our three other hypotheses have not been confirmed. The results only partially confirm our fourth hypothesis, as higher importance given to self-enhancement values

predicted a higher controlled motivation. This latter result confirms Schwartz's [1] and Maio's [2] expectations again and aligns with Adell et al. [24], who found that the more participants attached importance to self-enhancement values, the less they were intrinsically motivated to do sports in competition. Attaching importance to power and achievement values in a sport and exercise context may make individuals think they must be the best at every training session (i.e., dominating others) and feel bad when this does not happen. This kind of extrinsic motivation may lead individuals attaching high importance to power and achievement values, especially less-performing people, to give up the activity in the long run and to experience burnout [54].

The same may be true in the context of physical activity. We could speculate that individuals attaching high importance to power and achievement values should feel systematically powerful and dominant (e.g., running fast, not being tired) when doing physical activity to be motivated. Getting back into shape may be more difficult for self-enhancement-oriented individuals as they may feel easily dominated by their bodies and not like this feeling. They may also experience displeasing emotions when doing physical activity with others if they project a public image of someone not physically dominant.

Nevertheless, cultural factors may also explain the unexpected results of controlled motivation. There are solid external and normative pressures to exercise in China [37–39], and research has shown that social norms could mitigate the values-behaviors relationships [17]. Cultures strongly emphasizing individual autonomy, as in Western countries, encourage individuals to express their values, while cultures with rigorous norms, as in Eastern countries, restrict the expression of personal values [17,55]. This may explain in our research why only self-enhancement values are significantly related to controlled motivation and why controlled motivation did not mediate between values and physical activity.

## Limitations and future research

Only three studies have been conducted on the relationship between Schwartz's values model and self-determination theory [5,24,25]. One of these studies has been published in a Spanish journal with a limited sample [24]. Another study has been published in a book chapter using the Schwartz Value Survey among adolescents [2], although using the SVS with a young population is not advised. Consequently, our research is the first to study connections and interrelations between values, SDT, and self-reported physical activity.

However, a few limitations of this study and avenues for future research must be stated. Specifically, we used a cross-sectional design, which is inherently limited in establishing causality, as this methodology captures associations at a single time point without accounting for temporal dynamics. Consequently, further experimental or longitudinal research is needed to continue exploring the relationship between values, self-determined motivation, and physical activity. Such studies could elucidate whether changes in values and motivation precede or follow changes in physical activity and reveal potential reciprocal relationships, thereby informing more targeted interventions to promote sustained physical activity. Also, our research did not control students' socioeconomic status, and our data came from only one university in Shanghai, limiting our findings' generalizability. Socioeconomic status moderates the relationship between values and well-being in China [56]. An interesting perspective would be to test whether socioeconomic status moderates the relationship between values, self-determined motivation, and physical activity. Moreover, we focused on Chinese sports science students in the present research. An exciting issue should be to compare how values may be related to autonomous and controlled motivation in different cultures (e.g., Europe and Asia). Finally, we used a self-report measure of physical activity level. Participants with this kind of measure may inaccurately report the frequency, intensity, or duration of physical

activities or be influenced by social desirability bias to meet perceived expectations [57]. An important step in studying the relationship between values, motivation, and physical activity will be using the accelerometer [58] to measure physical activity levels objectively.

Understanding the factors that motivate or inhibit physical activity is essential concerning health issues (e.g., obesity). Values are important, highly abstract psychological constructs that may help to understand physical activity engagement. Specifically, understanding the relationship between values and specific motivation, like Self-Determined Motivation, is an interesting step in this direction.

## Supporting information

**S1 File. Separate mediation effects on autonomous, controlled, and a-motivation.** (DOCX)

**S1 Table. Definition and examples of separate motivational regulation in BREQ-2.** (DOCX)

## Author contributions

**Data curation:** Yan Liang.

**Funding acquisition:** Jian Yang.

**Methodology:** Yan Liang, Nicolas Souchon.

**Software:** Yan Liang.

**Supervision:** Olivier Rascle, Jian Yang, Nicolas Souchon.

**Validation:** Olivier Rascle, Jian Yang, Nicolas Souchon.

**Visualization:** Olivier Rascle, Nicolas Souchon.

**Writing – original draft:** Yan Liang, Nicolas Souchon.

**Writing – review & editing:** Olivier Rascle, Nicolas Souchon.

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
