## [Decision Letter · Decision Letter 0]

20 Nov 2024

PONE-D-24-49781Values, motivation, and physical activity among Chinese sport sciences studentsPLOS ONE

Dear Dr. Yang,

Thank you for submitting your manuscript to PLOS ONE. After careful consideration, we feel that it has merit but does not fully meet PLOS ONE’s publication criteria as it currently stands. Therefore, we invite you to submit a revised version of the manuscript that addresses the points raised during the review process.

We look forward to receiving your revised manuscript.

Kind regards,

Henri Tilga, PhD

Academic Editor

PLOS ONE

Journal Requirements:

6. Please include captions for your Supporting Information files at the end of your manuscript, and update any in-text citations to match accordingly. Please see our Supporting Information guidelines for more information: http://journals.plos.org/plosone/s/supporting-information .

Reviewers' comments:

Reviewer's Responses to Questions

**Comments to the Author**

1. Is the manuscript technically sound, and do the data support the conclusions?

Reviewer #1: Yes

2. Has the statistical analysis been performed appropriately and rigorously? 

Reviewer #1: Yes

3. Have the authors made all data underlying the findings in their manuscript fully available?

Reviewer #1: Yes

4. Is the manuscript presented in an intelligible fashion and written in standard English?

Reviewer #1: Yes

5. Review Comments to the Author

Reviewer #1: I would like to thank for the opportunity to review this manuscript. Please see the following comments to consider to further increase the quality of this manuscript.

The hypotheses (H1 to H4) are conceptually sound but could be presented more precisely. For example, some predictions are implicit rather than explicit, making it harder for readers to grasp specific expectations at a glance. Clearly state each hypothesis as either a direct or mediated relationship between values, motivation, and physical activity.

The literature review briefly covers previous findings but could benefit from more detailed discussions. For instance, explaining how values influence physical activity through SDT (autonomy, competence, and relatedness) would make the theoretical background more comprehensive.

Expand on previous findings with similar populations (e.g., sports science students or Chinese students) to underscore the relevance of the chosen sample.

Although the sample demographics (age, BMI) are included, the manuscript lacks information on the socioeconomic status or geographic distribution within China, which might impact the generalizability of the findings. Adding this data or discussing limitations regarding generalizability would enhance the robustness of the analysis.

The section describing measures could benefit from more precise information. For example, clarifying the specific aspects of the IPAQ short form used and why it was chosen over other physical activity measures could provide context.

More justification on the selection of control variables, especially BMI and competition level, would strengthen the methodological rigor. It would be helpful to discuss potential influences of these factors on motivation and physical activity.

The results reveal interesting but unexpected findings regarding controlled motivation, particularly concerning self-enhancement values. This aspect deserves more exploration in the discussion. Consider discussing potential cultural influences on controlled motivation in the Chinese context and contrasting with findings from Western contexts.

While limitations are briefly mentioned, additional discussion on limitations would add value. For example, address the limitations of cross-sectional design more fully and suggest how longitudinal studies could clarify the direction of causality between values, motivation, and physical activity.

Another limitation is the self-reported nature of physical activity, which can introduce bias. Consider discussing how objective measures (e.g., pedometers or accelerometers) could validate findings in future research.

6. PLOS authors have the option to publish the peer review history of their article (what does this mean? ). If published, this will include your full peer review and any attached files.

**Do you want your identity to be public for this peer review?** For information about this choice, including consent withdrawal, please see our Privacy Policy .

Reviewer #1: No

---

## [Author Response · Author response to Decision Letter 1]

14 Dec 2024

16 December 2024

PLOS ONE

Dear Editor Henri Tilga and the reviewer,

Thank you for inviting us to revise our manuscript, “Values, motivation, and physical activity among Chinese sport sciences students,” for potential publication in PLOS ONE. We greatly appreciate your detailed, constructive feedback.

This letter summarizes how our revision has responded to your points. We have attempted to address every point that was raised. Also, we resubmit our revised manuscript with tracked changes in yellow to facilitate the review. We hope you will view our revised manuscript as suitable for publication in the journal.

Responses to the Editor,

1. The editor’s first point is to ensure the manuscript meets PLOS ONE's style requirements, including those for file naming.

We want to thank the editor for the kind reminder. We modified all the documents according to PLOS ONE's requirements.

2. The editor’s second point is that strongly recommend all authors decide on a data sharing plan before acceptance, as the process can be lengthy and hold up publication timelines.

We thank you again for your reminder. Now, all the authors of this manuscript agree to share the raw data so that our entire data set can be freely accessible to all readers.

3. The editor’s third point that PLOS requires an ORCID iD for the corresponding author in Editorial Manager on papers submitted after December 6th, 2016. Ensure that have an ORCID iD and that it is validated in Editorial Manager.

We added the ORCID ID for our corresponding author, Jian Yang (0000-0003-4062-4646), in the re-submission process and authenticated the pre-existing ID in Editorial Manager.

4. The editor’s fourth point that our grant information in the ‘Funding Information’ and ‘Financial Disclosure’ sections do not match.

We greatly appreciate the reminder. Upon review, we noticed that the information in the 'Funding Information' and 'Financial Disclosure' sections did not match. We would like to corrected the information to ensure consistency. However, we could not find the 'Financial Disclosure' option in the re-submission system. It seems that the system may have automatically confirmed the absence of financial support based on our previous submission.

Our study was supported by the Humanities and Social Science Fund of the Ministry of Education of China (Award Number: 22YJA890032), and granted to Jian Yang, Ph.D. The funder, Jian Yang, was responsible for data collection in China.

We want to correct the “Financial Disclosure” section to: “This study was supported by the Humanities and Social Science Fund of Ministry of Education of China (Award Number: 22YJA890032), awarded to Jian Yang, Ph.D. The funder, Jian Yang, was responsible for data collection in China.”

We apologize for our unfamiliarity with the re-submission system. We kindly request the editor's help updating this information and sincerely appreciate the assistance in addressing this issue. Additionally, we have reported this issue to the journal staff, as we believe they may be more familiar with this process.

5. For the editor’s fifth point that include a separate caption for each figure in your manuscript.

We added captions for our figures in the manuscript. On p.4, line 77, we add “Fig 1. Schwartz’s model of values.” On p.16, line 327, we add “Fig 2. Tested model in the present study.”

6. The editor’s sixth point is to include captions for our Supporting Information files at the end of manuscript, and update any in-text citations to match accordingly.

We added the caption for our Supporting Information file at the end of our manuscript according to the Supporting Information guidelines, “S1 File. Separate mediation effects on autonomous, controlled, and a-motivation”. We also updated all the in-text citations with our revision. Please see the updated citations in the revised manuscript.

Responses to the reviewer,

1. The review’s first point is that the hypotheses (H1 to H4) are conceptually sound but could be presented more precisely. For example, some predictions are implicit rather than explicit, making it harder for readers to grasp specific expectations at a glance. Clearly state each hypothesis as either a direct or mediated relationship between values, motivation, and physical activity.

We agree with your point. Thank you very much for this feedback. Accordingly, we modified the four hypotheses into clearer descriptions. For example, we wrote for the first hypothesis p. 8, lines 188 to 193: “H1: Openness to change (Independent Variable, IV) and self-transcendence values (IV) would positively predict physical activity (Dependent Variable, DV), and autonomous motivation (Mediator, M) would mediate this relationship. Precisely, a greater attachment to openness to change (IV) and self-transcendence values (IV) would predict a greater autonomous motivation (M), which would predict a greater level of physical activity (DV)”. We hope this more precise way of writing the hypothesis makes it more straightforward.

2. The reviewer’s second point is that the literature review briefly covers previous findings but could benefit from more detailed discussions. For instance, explaining how values influence physical activity through SDT (autonomy, competence, and relatedness) would make the theoretical background more comprehensive.

Again, we agree with your feedback. Thank you very much. Accordingly, we add two paragraphs in the rationale to explain our theoretical expectations more precisely. We wrote p. 6, lines 121 to 128: “Schwartz et al. [3] theorize that growth/anxiety-free values: i.e., openness to change and self-transcendence, should be related to autonomous-intrinsic motivation. Specifically, openness to change encompasses stimulation and self-direction values related to psychological needs for mastery and variety. Self-transcendence encompasses universalism and benevolence values related to needs for affiliation [1-3]. Consequently, individuals prioritizing openness to change and self-transcendence values may focus on satisfying their competence, autonomy, and affiliation needs, while research has shown that satisfying these needs increases autonomous motivation in SDT [4, 13, 22].”

Also, we wrote p. 6, lines 129 to 135: “Moreover, self-protection/anxiety-avoidance values: i.e., self-enhancement and conservation, should be related to controlled-extrinsic motivation as self-enhancement values are related to extrinsic needs such as wealth and social recognition (power and part of achievement). Also, conservation values are related to following rules, avoiding social violations (conformity, tradition), or getting protection and attention (security) [1-3]. All these extrinsic needs related to conservation values tend to increase extrinsic motivation in SDT [4, 13, 23].” We hope these more detailed explanations make the theoretical part more understandable.

3. The reviewer’s third point is that expand on previous findings with similar populations (e.g., sports science students or Chinese students) to underscore the relevance of the chosen sample.

We agree perfectly with your point. Thank you again for this important feedback. We now explain specifically why we focused on a sample of sports sciences students in China. We wrote p. 7 and 8, lines 164 to 175: “To our knowledge, numerous studies have been conducted to understand self-determined motivation toward physical activity among general university students in Europe [31, 32] or in China [33, 34]. Still, very few studies have been undertaken to understand physical activity among sports sciences students [5, 35]. Research has shown sports science students doing higher physical activity in Western Europe compared to the general exerciser [35], suggesting sports sciences students pursuing a deep autonomous motivation to do physical activity in this individualistic culture [36]. In contrast, in the Eastern collectivist cultural context, particularly in China, motivation to exercise among general students is heavily influenced by external pressures like norms, institutional policies, and parental and coach expectations [37, 38, 39]. All in all, this makes the choice to study the relation between values, autonomous motivation, and physical activity among sports sciences students in China interesting”.

4. The reviewer’s fourth point is that although the sample demographics (age, BMI) are included, the manuscript lacks information on the socioeconomic status or geographic distribution within China, which might impact the generalizability of the findings. Adding this data or discussing limitations regarding generalizability would enhance the robustness of the analysis.

We want to thank you again for this constructive feedback. Yes, socioeconomic status (SES) and geographic variables are vital factors in physical activity research. For example, different studies made with Chinese samples indicated that children, adolescents [1] (reference in the response letter), and college students [2] (reference in the response letter) with higher levels of SES in China can more easily engage in physical activity as they can pay for sporting clubs and physical activity programs in gym clubs.

We did not find data or research on students' socioeconomic origins at East China Normal University. These data may not be communicated in China. Consequently, we add within the limitation part of the research: p. 20, lines 413 to 417 “Also, our research did not control students’ socioeconomic status, and our data came from only one university in Shanghai, limiting our findings' generalizability. Socioeconomic status moderates the relationship between values and well-being in China [56]. An interesting perspective would be to test whether socioeconomic status moderates the relationship between values, self-determined motivation, and physical activity”.

5. The reviewer’s fifth point is that the section describing measures could benefit from more precise information. For example, clarifying the specific aspects of the IPAQ short form used and why it was chosen over other physical activity measures could provide context.

We thank you very much for this feedback. Our study employed one questionnaire to assess physical activity (PA), another for value, and the last for measuring motivation to exercise. These three questionnaires could take a very long time to complete. We carefully consider minimizing the assessment duration to maintain participants' focus and ensure accurate responses. To achieve this, the short version of the International Physical Activity Questionnaire (IPAQ-SF), which consisted of 7 items for PA, was selected as it takes a short time to complete and was well-validated cross-culturally, including the Chinese context. Finally, participants required approximately 12 to 20 minutes to complete all the questionnaires.

Accordingly, we explain in p.11, lines 247 to 274: “We needed to shorten our time measure to increase the number of participants agreeing to do our research. To this end, we used the International Physical Activity Questionnaire short form [21]. The IPAQ-SF is widely used to assess global physical activity and has been extensively validated across cultural contexts [44, 45]. Notably, the seven-item short version of the IPAQ has been typically recommended as a cost-effective method to access global PA [44]. Its brevity minimizes the burden on participants, which is particularly advantageous in studies involving multiple measurements. Above all, we employed the well-validated Chinese version of the IPAQ-SF [46].

The IPAQ short assesses physical activity undertaken across a comprehensive set of domains in daily life through 7 items and three different intensities in global physical activity. The first question for measuring vigorous physical activity is: “During the last 7 days, how many days did you do vigorous physical activities like heavy lifting, digging, aerobics, or fast bicycling?”. The second question for measuring moderate physical activity is: “During the last 7 days, how many days did you do moderate physical activity like carrying light loads, bicycling at a regular pace, or doubles tennis? Do not include walking”. The third question for measuring “walking” is: “Think about the time you spent walking in the last 7 days. This includes at work and home, walking to travel from place to place, and any other walking you have done solely for recreation, sport, exercise, or leisure”. For each of these three questions, participants answered the frequency in the last 7 days of vigorous and moderate physical or walking activities and further answered the time they spent on one of those days.

The outcome is a continuous score reported as the median Metabolic Equivalent of Task (MET)-minutes, which express the energy cost of an activity. Median MET values can be calculated for walking (W), moderate-intensity activities (M), and vigorous-intensity activities (V) using the subsequent methods: walking MET-minutes/week=3.3*walking minutes*walking days; moderate MET-minutes/week=4.0*moderate-intensity activity minutes*days; vigorous MET-minutes/week=8.0*vigorous-intensity activity minutes*days. A combined “total” physical activity MET-min/week can be calculated as the aggregate of walking + Moderate + Vigorous MET-min/week scores”.

6. The reviewer’s sixth point is that more justification on the selection of control variables, especially BMI and competition level, would strengthen the methodological rigor. It would be helpful to discuss potential influences of these factors on motivation and physical activity.

We agree with this very constructive feedback. We developed accordingly the control variable part. We wrote p.12, lines 284 to 294: “Participants indicated their age, gender, Body Mass Index (BMI =Kg/m²) through their weight (in kilograms) and height (in meters), and competition level (no competition, local, regional, national, or international) as control variables. Specifically, research has shown that students with higher BMIs report lower levels of autonomous motivation for engaging in physical activities [50], body image concerns and discomfort during exercise [51], or a lower level of endurance during physical activity tasks [52]. Moreover, research has shown that sporting competition levels impact physical activity, as students who do a sport in competition tend to be more physically active than students doing physical activity without sporting competition [8]. Different external pressures (e.g., coaches, head of the sporting club) may be directed toward students doing sports in competition, raising their level of controlled motivation [25]”.

7. The reviewer’s seventh point is the results reveal interesting but unexpected findings regarding controlled motivation, particularly concerning self-enhancement values. This aspect deserves more exploration in the discussion. Consider discussing potential cultural influences on controlled motivation in the Chinese context and contrasting with findings from Western contexts.

Thank you very much for this constructive feedback. We added to the discussion p. 19, lines 390 to 397: “Nevertheless, cultural factors may also explain the unexpected results of controlled motivation. There are solid external and normative pressures to exercise in China [37, 38, 39], and research has shown that social norms could mitigate the values-behaviors relationships [17]. Cultures strongly emphasizing individual autonomy, as in Western countries, encourage individuals to express their values, while cultures with rigorous norms, as in Eastern countries, restrict the expression of personal values [17, 55]. This may explain in our research why only self-enhancement values are significantly related to controlled motivation and why controlled motivation did not mediate between values and physical activity”.

8. The reviewer’s last point is while limitations are briefly mentioned

---

## [Decision Letter · Decision Letter 1]

17 Dec 2024

Values, motivation, and physical activity among Chinese sport sciences students

PONE-D-24-49781R1

Dear Dr. Yang,

We’re pleased to inform you that your manuscript has been judged scientifically suitable for publication and will be formally accepted for publication once it meets all outstanding technical requirements.

Kind regards,

Henri Tilga, PhD

Academic Editor

PLOS ONE

Additional Editor Comments (optional):

Reviewers' comments:

Reviewer's Responses to Questions

**Comments to the Author**

1. If the authors have adequately addressed your comments raised in a previous round of review and you feel that this manuscript is now acceptable for publication, you may indicate that here to bypass the “Comments to the Author” section, enter your conflict of interest statement in the “Confidential to Editor” section, and submit your "Accept" recommendation.

Reviewer #1: All comments have been addressed

2. Is the manuscript technically sound, and do the data support the conclusions?

Reviewer #1: Yes

3. Has the statistical analysis been performed appropriately and rigorously? 

Reviewer #1: Yes

4. Have the authors made all data underlying the findings in their manuscript fully available?

Reviewer #1: Yes

5. Is the manuscript presented in an intelligible fashion and written in standard English?

Reviewer #1: Yes

6. Review Comments to the Author

Reviewer #1: Authors have done well job on revising their manuscript. I think this manuscript is ready for the publication.

7. PLOS authors have the option to publish the peer review history of their article (what does this mean? ). If published, this will include your full peer review and any attached files.

**Do you want your identity to be public for this peer review?** For information about this choice, including consent withdrawal, please see our Privacy Policy .

Reviewer #1: No

---

## [Editor Report · Acceptance letter]

PONE-D-24-49781R1

PLOS ONE

Dear Dr. Yang,

I'm pleased to inform you that your manuscript has been deemed suitable for publication in PLOS ONE. Congratulations! Your manuscript is now being handed over to our production team.

Kind regards,

on behalf of

Dr. Henri Tilga

Academic Editor

PLOS ONE